# Electromagnetic Spectrum Allocation Method for Multi-Service Irregular Frequency-Using Devices in the Space–Air–Ground Integrated Network

**DOI:** 10.3390/s22239227

**Published:** 2022-11-27

**Authors:** Yongchao Meng, Peihan Qi, Qian Lei, Zhengyu Zhang, Jinyang Ren, Xiaoyu Zhou

**Affiliations:** 1State Key Laboratory of Integrated Service Networks, Xidian University, Xi’an 710071, China; 2Northwest Air Traffic Management Bureau, Civil Aviation Administration of China, Xi’an 710082, China

**Keywords:** space–air–ground integrated network, spectrum allocation, priority, orthogonal experiment, niche method, micro genetic algorithm

## Abstract

The management and allocation of electromagnetic spectrum resources is the inner driving force of the construction of the space–air–ground integrated network. Existing spectrum allocation methods are difficult to adapt to the scenario where the working bandwidth of multi-service frequency-using devices is irregular and the working priorities are different. In this paper, an orthogonal genetic algorithm based on the idea of mixed niches is proposed to transform the problem of frequency allocation into the optimization problem of minimizing the electromagnetic interference between frequency-using devices in the integrated network. At the same time, a system model is constructed that takes the minimum interference effect of low-priority-to-high-priority devices as the objective function and takes the protection frequency and natural frequency as the constraint conditions. In this paper, we not only introduce the thought of niches to improve the diversity of the population but also use an orthogonal uniform crossover operator to improve the search efficiency. At the same time, we use a standard genetic algorithm and a micro genetic algorithm to optimize the model. The global searchability and local search precision of the proposed algorithm are all improved. Simulation results show that compared with the existing methods, the proposed algorithm has the advantages of fast convergence, strong stability and good optimization effect.

## 1. Introduction

The deployment and application of 5th-generation mobile communication (5G) has opened the era of the Internet of everything and promoted the integration of the information industry and other traditional industries [1]. With the commercialization of 5G technology, many countries and organizations have made forward-looking layouts for 6th-generation communication (6G) [2,3]. In order to support full coverage of the network and the high-speed mobility of users, the breadth and depth of the existing communication range of the ground network are extended. The 6G technology will cover network infrastructure such as airspace and space and integrate ground and non-ground networks to provide full coverage of space, sky and land [4,5].

The space–air–ground integrated network integrates space-based networks of orbiting satellites, high-attitude platforms (HAPs) and HAPs consisting of unmanned aerial vehicles (UAVs) [6,7]. The ground-based network, composed of traditional wireless communication systems, can achieve seamless coverage of the whole domain and provide users with ubiquitous communication services, which is the core development direction of 6G technology in the future [8]. At present, the research on the space–air–ground integrated network mainly includes joint beamforming, power allocation and integration with the Internet of Things (IoT). In [9], the authors proposed a joint optimization design for a non-orthogonal multiple access (NOMA)-based satellite–terrestrial integrated network (STIN), where a satellite multicast communication network shares the millimeter wave spectrum with a cellular network employing NOMA technology. In [10], the authors investigated the multicast communication of a satellite and aerial integrated network (SAIN) with rate-splitting multiple access (RSMA) to satisfy the explosive access demand of IoT devices. However, due to the differences of services among space-based networks and ground-based networks, the heterogeneity of frequency-using devices and the diversity of management, the space–air–ground integrated network is facing great challenges in the management and allocation of electromagnetic spectrum resources [11,12].

Spectrum allocation technology has been of great interest to researchers at home and abroad. In general, spectrum allocation technologies can be divided into the following three categories: Firstly, there are spectrum allocation technologies with regions as allocation objects [13,14]. This technology takes “cell” as the frequency object and the interference between cells as the constraint condition. Genetic algorithms, particle swarm optimization algorithms and other heuristic algorithms are used to achieve the frequency demand of each cell. Such methods solve the spectrum allocation problem to a certain extent but do not specifically consider the resource requirements of specific devices. Thus, they cannot be extended to the actual problem of different types of devices having different resource requirements and different frequency priorities. Secondly, there is spectrum allocation technology based on the frequency-using device as the allocation object [15,16]. This technology takes specific frequency-using device as the spectrum allocation objects, takes protection frequencies and prohibition frequencies as constraints, and takes same frequency interference and adjacent frequency interference between devices as optimization objectives to model. At the same time, an improved heuristic algorithm is used to solve the problem. Such algorithms achieve fine spectrum allocation at the device level, but the modeling of the spectrum allocation problem does not meet the actual business needs and does not take into account the difference in frequency priority. Thirdly, there is spectrum allocation or access technology based on cognitive radio or dynamic spectrum access [17,18]. This technology divides frequency-using device into two categories—primary users and secondary users—and allocates resources to secondary users on the premise of ensuring the performance requirements of primary users. The constraints are the maximum interference threshold in the same channel and the maximum transmission power of each secondary user. The optimization goal is to maximize the spectral efficiency of each user or the number of users successfully allocated. Reinforcement learning or heuristic algorithms are usually used to solve the problem. These algorithms can adapt to the dynamic electromagnetic environment and distributed decision-making scenarios and have good flexibility and robustness. However, the problem of differentiation of frequency demands of users at all levels has not been solved, and the method of dividing users at all levels cannot adapt to scenarios with different priorities.

Generally speaking, the basic thought behind existing technology is similar: abstracting spectrum resources into several resource blocks with different properties while taking the electromagnetic interference between devices, device transmission power or various frequency restrictions as constraints. The purpose is to minimize the interference to the devices and meet the frequency requirements of each device. At the same time, various algorithms are used to solve the electromagnetic spectral distribution problem to a certain extent. However, the space–air–ground integrated network is a huge and complex system; it is a highly integrated system of multi-dimensional networks. The frequency requirements of different business types of devices vary greatly, and the existing technology cannot meet the irregular frequency requirements of various business types of devices in complex networks. In addition, in the actual business scenario, each frequency-using device performs different types of business, and different business types have different contributions and impacts to the network. Therefore, frequency requirements for devices providing high-priority services should be preferred. Existing techniques do not consider the practical problem of different frequency-using priorities for each device. For the above problems, this paper firstly conducts mathematical modeling on the spectrum allocation problem of frequency-using devices in the multi-service irregular scenario and proposes a multi-service irregular frequency allocation method based on mixed niche orthogonal genetic algorithms. The proposed algorithm introduces niche thoughts based on the clearing mechanism to improve the diversity of the population and proposes orthogonal uniform crossing operators to improve search efficiency. Moreover, the proposed algorithm uses a standard genetic algorithm (SGA) and a micro genetic algorithm (MGA) to improve the global search capability and local search accuracy.

## 2. System Model

We show the spectrum allocation system model of frequency-using devices with different operating frequencies and priorities in the space–air–ground integrated network in Figure 1. The system model consists of a scene description, constraint conditions and objective functions. When a large number of frequency-using devices apply for working frequency bands from the spectrum management system, the spectrum management system will allocate spectrum according to the natural frequency band of each device, its priority, the system’s rejection frequency band, and the interference between devices.

Assuming that there are *N* devices in the system, the differences between devices of different business types are mainly reflected in the parameter numerical level, and device of different business types can be described in *k* unified ways. This paper specifically describes a device *k* using the parameters shown in Table 1, where $p_{k}$ is the value of device priority. The higher the priority, the smaller the value of $p_{k}$. The variables $x_{k}$, $y_{k}$ and $z_{k}$, respectively, represent the longitude, latitude and height of the device deployment point; $f_{kgs}$ and $f_{kge}$ represent the start and end frequencies of the device’s natural frequency band, respectively; $f_{ks}$ and $f_{ke}$ represent the start and end frequencies of the device’s working frequency band, respectively; the device’s working frequency band must be within the range of the natural frequency band.

### 2.1. Constraint Conditions

According to the scenario description, the spectrum allocation scheme must meet the following three constraints:The operating frequency band of each device shall be within its inherent frequency band range, as shown in Equation (Equation 1):
(1)$$\left|\frac{f_{ks}+f_{ke}}{2}-\frac{f_{kgs}+f_{kge}}{2}\right|+\frac{B_{k}}{2}\leq \frac{f_{kge}-f_{kgs}}{2}$$The operating frequency band of each device should not overlap with the protection frequency band of its protection area, as shown in Equation (Equation 2):
(2)$$\left|\frac{f_{ks}+f_{ke}}{2}-\frac{f_{kps}+f_{kpe}}{2}\right|\geq +\frac{B_{k}+\left(f_{kge}-f_{kgs}\right)}{2}$$The operating frequency band of each device shall not overlap with any rejection frequency band, as shown in Equation (Equation 3):
(3)$$\left|\frac{f_{ks}+f_{ke}}{2}-\frac{f_{nas}+f_{nae}}{2}\right|\geq +\frac{B_{k}+\left(f_{nae}-f_{nas}\right)}{2}$$

### 2.2. Objective Function

The objective of the model is to avoid interference with or reduce interference to high-priority devices. Interference can be considered from two aspects:

Firstly, frequency domain interference: if the working frequency bands do not overlap, then there is no frequency domain interference between the two devices. If there is overlap, the severity of interference is described by the frequency spectrum overlap degree. Based on this, the frequency domain interference coefficient is defined as shown in Equation (Equation 4): (4)$$Q_{ij}\left(f_{ic},f_{jc}\right)=\left\{\begin{array}{cc} \frac{\left(B_{i}+B_{j}\right)/2-abs\left(f_{ic}-f_{jc}\right)}{B_{j}} & abs\left(f_{ic}-f_{jc}\right)<\frac{\left(B_{i}+B_{j}\right)}{2}\\ 0 & abs\left(f_{ic}-f_{jc}\right)\geq \frac{\left(B_{i}+B_{j}\right)}{2} \end{array}\right.$$
where $f_{ic}=0.5\cdot \left(f_{is}+f_{ie}\right)$ is the center frequency of the operating frequency band of the *i*-*th* device, and i,j∈[1,N].

Secondly, airspace conflict: assuming that the transmitter device sends signals with power $P_{i}$ and the receiver device sensitivity is $S_{j}$, in the process of signal propagation, path loss $L_{ij}$ is generated, as shown in Equation (Equation 5) [19]: (5)$$L_{ij}\left(f_{ic}\right)=32.44+20\log_{10}f_{ic}+20\log_{10}D_{ij}$$
where $D_{ij}$ represents the straight-line distance between the *i-th* device and the *j-th* device, and i,j∈[1,N],i≠j. The spatial interference coefficient is defined to quantify spatial interference, as shown in Equation (Equation 6): (6)$$I_{ij}\left(f_{ic}\right)=\left\{\begin{array}{cc} 1 & P_{i}-L\left(f_{ic}\right)\geq S_{j}\\ 0 & P_{i}-L\left(f_{ic}\right)\leq S_{j} \end{array}\right.$$

When $I(f_{ic})=1$, there is spatial interference between the originating device and the receiving device. Otherwise, there is no airspace interference.

The device interference coefficient $a_{i}\left(F_{c}\right)$ is defined to quantify the degree that each device does not interfere with other devices, as shown in Equation (Equation 7): (7)$$a_{i}\left(F_{c}\right)=\sum_{j=1,j\neq i}^{N}[\left(1-Q_{ij}\left(f_{ic},f_{jc}\right)\right)\cdot \left(1-I_{ij}\left(f_{ic}\right)\right)\cdot \left(p_{\max}-p_{j}\right)],i\in \left[1,N\right]$$
where $F_{c}=\left\{f_{1c},f_{2c},\ldots ,f_{Nc}\right\}$,$p_{\max}$ is the priority value with the largest absolute value, that is, the lowest priority value.

The fitness function $S(F_{c})$ is defined as the objective function of the spectrum allocation model, as shown in Equation (Equation 8): (8)$$S\left(F_{c}\right)=\sum_{i=1}^{N}a_{i}\left(F_{c}\right)=\sum_{i=0}^{N}\sum_{j=1,j\neq i}^{N}[\left(1-Q_{ij}\left(f_{ic},f_{jc}\right)\right)\cdot \left(1-I_{ij}\left(f_{ic}\right)\right)\cdot \left(p_{\max}-p_{j}\right)]$$

The optimization objective of the model is to find the appropriate working frequency band for each device so that the interference of low-priority-to-high-priority devices is minimized.

In conclusion, the spectrum allocation model as shown in Equation (Equation 9) is established: (9)$$\begin{array}{c} \begin{array}{cc} \max_{\left\{F_{c}\right\}} & S(F_{c}) \end{array},\\ \begin{array}{cc} s.t. & abs\left(\frac{f_{ks}+f_{ke}}{2}-\frac{f_{kgs}+f_{kge}}{2}\right)+\frac{B_{k}}{2}\leq \frac{\left(f_{kge}-f_{kgs}\right)}{2},\\ & abs\left(\frac{f_{ks}+f_{ke}}{2}-\frac{f_{kps}+f_{kpe}}{2}\right)\geq \frac{B_{k}+\left(f_{kpe}-f_{kps}\right)}{2},\\ & abs\left(\frac{f_{ks}+f_{ke}}{2}-\frac{f_{nas}+f_{nae}}{2}\right)\geq \frac{B_{k}+\left(f_{nae}-f_{nas}\right)}{2},\\ & k\in [1,N],n\in [1,A]. \end{array} \end{array}$$

## 3. Working Process of the Model

### 3.1. Description of the Algorithm

In this paper, a method is proposed to solve the multi-service irregular spectrum allocation problem. Firstly, several spectrum allocation schemes that meet the constraints are randomly generated. The content of the spectrum allocation scheme is the operating frequency band information of each device. Then, each spectrum allocation scheme is encoded with real numbers, and the scheme is mapped into a chromosome array. Each chromosome array is an individual, and the fitness function to evaluate the performance of each individual is the objective function of the spectrum allocation model. Randomly generated individuals form the initial population, which is used as the starting point for the genetic algorithm iteration. In terms of solving the algorithm, this paper proposes a new Hybrid Niche Orthogonal Genetic Algorithm (HNOGA). The steps of the HNOGA algorithm are shown in the pseudocode of Algorithm 1.
**Algorithm 1** HNOGA pseudocode1:**Input:** population size *M*, number of elite individuals *q*, subpopulation neighborhood search probability $P_{m1}$, optimal individual neighborhood search probability $P_{m2}$, and maximum iteration number *T*;2:**Output:** the optimal individual $I_{b}$;3:**Start:**4:Step 1: The initial population with population size *M* was generated by random coding, the fitness of each individual in the initial population was calculated, and the top *q* individuals with the highest fitness were copied and saved $\left(q<M\right)$. Studies show that when q=0.3M, the performance of the algorithm is optimal.5:Step 2: Genetic algorithm optimization6:**for** iteration t=1:T
**do**7:    Main genetic algorithm operation: random league selection, orthogonal uniform crossover, uniform mutation;8:     All the individuals whose fitness is not 0 in the population and whose subpopulation contains greater than or equal to 2 individuals are optimized by the micro genetic algorithm with probability $P_{m1}$, and then the niche operation is performed;9:     The individual with the highest fitness in the whole population was selected to perform micro genetic algorithm optimization with probability $P_{m2}$;10:    Perform niche operations;11:    The top *q* individuals with the highest fitness in the population are copied and retained, and the top *M* individuals with the highest fitness are used to form the population of the next iteration.12:Step 3: Output the optimal individual $I_{b}$ of the last generation population;13:**End:** Obtain the optimal individual $I_{b}$.

In each iteration, a standard genetic algorithm (SGA) is first used for global optimization (this step is referred to as the “main genetic algorithm” in this paper), and then the optimized population is niche operated: Firstly, the normalized Euclidean distance between individuals in the population is calculated. If the normalized Euclidean distance of some individuals is less than a certain threshold, they are clustered into a subpopulation, and all individuals in each subpopulation have similar genetic characteristics. Secondly, the individual with the highest fitness in the subpopulation retains its original fitness value, and the fitness of the other individuals is set to 0. This operation makes the probability that the genetic characteristics of each subpopulation will be eliminated in the selection process of the next iteration very low to ensure the genetic diversity of the population. After the niche operation, the original population is divided into several subgroups. Then, the individual with the highest fitness is selected from each subpopulation with population size greater than 1, and the micro genetic algorithm is used for a neighborhood search. Thus, the full utilization of neighborhood information is realized, and the search accuracy and efficiency of the algorithm are improved.

The flowchart of the HNOGA algorithm is shown in Figure 2.

### 3.2. Master Genetic Algorithm

#### 3.2.1. Coding and Initial Population Generation

The HNOGA algorithm uses real coding to map from the spectrum allocation scheme to the chromosome array. Assuming that there are *N* devices in the region, the chromosome is an array of real numbers $G_{s}=\left\{g_{1},g_{2},\cdots ,g_{N}\right\}$ of length *N*. Each gene locus $g_{k}$ of the chromosome corresponds to a device, and the value of $g_{k}$ is the central frequency of the operating band of the corresponding device. Assuming that the working bandwidth of the *k-th* device is $B_{k}$, the start frequency of the working band of the device is $f_{ks}=g_{k}-0.5B_{k}$ and the end frequency is $f_{ks}=g_{k}+0.5B_{k},k\in \left[1,N\right]$.

The HNOGA algorithm uses random coding to generate the initial population: for each device, the center frequency is randomly generated within its natural frequency band range, and the value is taken as the gene value of the corresponding position of the device on the chromosome. A complete individual can be generated by performing the above operations for all device. It should be noted that if the randomly generated center frequency value does not meet the three constraints of the model, it should be generated again until the location meets the constraints. Performing the above group size operation times generates the initial group.

#### 3.2.2. Function of Fitness

The fitness function of the HNOGA algorithm is the same as the objective function of the system model. The fitness $f(G_{s})$ of an individual $G_{s}=\left\{g_{1},g_{2},\ldots ,g_{N}\right\}$ is defined as shown in Equation (Equation 10): (10)$$f\left(G_{s}\right)=\sum_{i=1}^{N}a_{i}\left(G_{s}\right)=\sum_{i=0}^{N}\sum_{j=1,j\neq i}^{N}[\left(1-Q\left(g_{i},g_{j}\right)\right)\cdot \left(1-I\left(g_{i}\right)\right)\cdot \left(p_{\max}-p_{j}\right)]$$

#### 3.2.3. Selection Operator

The choice operator of the HNOGA algorithm is stochastic tournament selection. Compared with the commonly used roulette selection, the tournament selection strategy has the advantages of high solution accuracy and fast solution speed, so it is widely used in the construction of genetic algorithms [20,21]. The basic steps of the tournament selection strategy are as follows:

(1) total of *t* individuals are randomly selected from the parental population;

(2) Among the *t* individuals, the individuals with the highest fitness are selected and retained in the middle group;

(3) assuming that the parental population size is *M*, the above steps are repeated M−1 times to form a complete intermediate population.

The variable *t* is an artificially set parameter whose value greatly affects the actual effect of tournament selection. In this algorithm, the value *t* is set to 2.

#### 3.2.4. Crossover Operator

In this paper, a uniform crossover operator based on an orthogonal experiment is proposed; it is called an orthogonal uniform crossover operator.

Orthogonal experimental design is an efficient and economical experimental design method that mainly studies the influence mechanism of several specific factors in a system at different levels on the overall state or performance of the system. It designs a variety of the most-representative test schemes based on the principle of orthogonality to evenly disperse and match the levels of factors with the factors. The effect of a few tests is equivalent to that of a full test. The main tool for orthogonal experimental design is the orthogonal table, which is a matrix arranged by rows and columns and is usually represented as $L_{x}\left(q^{y}\right)$, where *L* represents the orthogonal table, *x* represents the number of trials, *q* represents the number of factor levels, and *y* represents the number of factors. This paper uses a two-level orthogonal table.

The orthogonal uniform crossover operator is an improvement and supplement to the uniform crossover operator, and it uses a similar operation mode as the uniform crossover operator. Each gene of the progeny chromosome is derived from one of the alleles at the corresponding location of the two parental chromosomes that have been paired [22,23]. The difference is that the uniform crossover operator randomly selects crossed genes to obtain two individuals, while the orthogonal uniform crossover operator quickly selects the best combination of alleles of parental individuals through orthogonal experiments to obtain only one individual. Specifically, the process of finding the best combination of alleles can be regarded as a two-level orthogonal experiment in which the factors are the genes on the chromosome and the two-level of each factor is the specific values of the corresponding alleles of the two parental chromosomes. According to the chromosome length (i.e., the number of factors), a two-level orthogonal table can be established for the experiment. Each experiment gets a new individual. The fitness of all individuals obtained by the orthogonal experiment is calculated, and the individual with the highest fitness is taken as the result of the orthogonal uniform crossover.

#### 3.2.5. Mutation Operator

The mutation operator of the HNOGA algorithm uses a uniform mutation operator, and the specific operation steps are as follows:

(1) Each locus of an individual is designated as the point of variation.

(2) For each gene locus, a uniform mutation operation is performed with mutation probability $p_{m}$.

(3) It is determined whether the working frequency band meets the three constraints of the model. If it does, continue; otherwise, go back to the second step.

(4) All individuals perform the above operations.

Compared with the variation strategy, the uniform variation strategy has greater variation intensity and better searchability. It can not only improve the diversity of the population, but also avoid the prematurity of the algorithm, which is suitable for solving complex problems.

### 3.3. Niche Operation

The HNOGA algorithm uses a niche technology based on a Clearing Procedure. Its basic idea is to classify all individuals based on the Euclidean distance between individuals, divide the whole population into multiple subgroups, and select the individual with the highest fitness within each subgroup to participate in further optimization [24,25]. The steps are as follows:

Step 1: If the fitness of two individuals is not 0, the normalized Euclidean distance between the two individuals is calculated. For any two individuals $G_{i}=\left\{g_{i1},g_{i2},\cdots ,g_{iN}\right\}$, $G_{j}=\left\{g_{j1},g_{j2},\cdots ,g_{jN}\right\}$, their normalized Euclidean distance is defined by Equation (Equation 11): (11)$$∥G_{i}-G_{j}∥=\sqrt{\frac{1}{N}\sum_{k=1}^{N}{\left(g_{ik}-g_{jk}\right)}^{2}},\left(\begin{array}{c} i=1,2,\ldots ,M-1\\ j=i+1,\ldots ,M \end{array}\right)$$

Step 2: For any two individuals, if their normalized Euclidean distance is less than or equal to the niche radius *D*, the fitness of the two individuals is compared. The fitness of the individuals with small fitness is set to 0, while the fitness of the individuals with large fitness remains unchanged and they are placed in a subgroup. If their normalized Euclidean distance is greater than the niche radius *D*, no operation is performed.

Step 3: All individuals perform the above operations.

These non-zero fitness individuals are essentially representatives of a class of individuals in the original population. Niche operations keep them in place in the iteration process, retaining the genetic characteristics of such individuals in the population and thus maintaining the diversity of the population. At the same time, this also avoids the phenomenon of a high-performance individual quickly replacing other individuals in the iterative process, resulting in the algorithm falling into a local optimum.

From the perspective of mathematical optimization, if the objective function of the problem solved by the genetic algorithm is a complex multi-peak function, the clustering phenomenon of some individuals in the population may be caused by the fact that these individuals search the neighborhood of the same local optimal solution. In the real genetic algorithm population, a class of individuals classified by Euclidean distance may be located near the same extreme point. The non-zero-fitness individual is the one closest to the local optimal solution after the niche operation. Then, a local search with higher precision can be further approached or even found with this individual as the starting point. Every extreme point of a multimodal function may be the global optimal solution, so the idea of a local search based on the individuals near the extreme point is significant for determining the global optimal solution. Local search methods are often referred to as “hill-climbing operators”.

### 3.4. Micro Genetic Algorithm

In this paper, MGA is extended and a real micro genetic algorithm is proposed [26,27]. The length of the chromosome $G_{m}=\left\{g_{p1},g_{p2},\cdots ,g_{pN}\right\}$ is equal to the number of devices *N*, and each gene location of the chromosome $g_{pk}$ corresponds to one device. The value $g_{pk}$ is the perturbation momentum of the center frequency of the operating frequency band of the corresponding device. The perturbation momentum should meet $g_{pk}\in \left[-g_{pb},g_{pb}\right]$, where $g_{pb}$ is the perturbation amplitude, namely the maximum perturbation momentum allowed by the algorithm $k\in \left[1,N\right]$. The micro genetic algorithm randomly generates the initial population in the same way as the main genetic algorithm.

Assuming that the optimization result of the main genetic algorithm is $G_{s}=$$\left\{g_{s1},g_{s2},\cdots ,g_{sN}\right\}$ and an individual of MGA is $G_{p}=\left\{g_{p1},g_{p2},\cdots ,g_{pN}\right\}$, then $G_{s}$ will generate a new individual *G* under the action of $G_{p}$. The action mechanism is shown in Equation (Equation 12): (12)$$G=G_{s}+G_{p}=\left\{g_{1},g_{2},\ldots ,g_{N}\right\}=\left\{g_{s1}+g_{p1},g_{s2}+g_{p2},\cdots ,g_{sN}+g_{pN}\right\}$$

The objective of the micro genetic algorithm is to find an appropriate perturbation $g_{s}$ to obtain a new individual *G* with the highest performance possible. Therefore, the objective function of the micro genetic algorithm is shown in Equation (Equation 13): (13)$$f\left(G\right)=\sum_{i=1}^{N}a_{i}\left(G\right)=\sum_{i=0}^{N}\sum_{j=1,j\neq i}^{N}\left[\left(1-Q\left(g_{i},g_{j}\right)\right)\cdot \left(1-I\left(g_{i}\right)\right)\cdot \left(p_{\max}-p_{j}\right)\right]$$

MGA used in this paper adopts three genetic operators, namely random league selection, orthogonal uniform crossover, and uniform mutation. It should be noted that MGA may produce illegal individuals in the process of coding and mutation, and $G_{s}$ will generate new individuals that do not meet the constraints of the system model under the action of the individual. In this paper, the method to deal with this problem is to re-encode or mutate until a legitimate individual is generated.

Through the combination of the main genetic algorithm and MGA, the HNOGA algorithm forms an iterative process including an outer loop and inner loop. When the number of outer loops is too many and the number of inner loops is too few, the local searchability of the algorithm decreases. When the number of outer loops is too small and the number of inner loops is too large, the global optimization ability of the algorithm is limited. Therefore, the ratio of internal and external circulation times should be reasonably allocated. The literature [28] puts forward that the optimal ratio of internal to external circulation is 0.3 to 0.5. In addition, an elite strategy is also used in the MGA, in which the worst individual in each iteration is replaced by the best individual in the previous iteration.

## 4. Simulation Results

### 4.1. Experimental Data Setting

Assume that there are 10 devices in the system, and the working bandwidth of each device is different. The data for each device are shown in Table 2.

In the actual scenario, there is only one protected zone. The start frequency and end frequency of the protected band in this zone are 150 MHz and 175 MHz, respectively. The protected area is a cuboid area with a latitude between 20 and 60 degrees, a longitude between 1 and 4 degrees, and a height between 20 and 50 m. There is only one denial spectrum in actual service scenarios. The rejection frequency band starts at 25 MHz and ends at 50 MHz. It can be seen that the total bandwidth of the commonly available frequency band is 200 MHz, while the sum of the working bandwidth of each device is 224 MHz. Since this paper does not consider the case of allocation failure, interference between devices cannot be avoided.

### 4.2. Simulation of Orthogonal Uniform Crossover Operator

To verify the performance of the orthogonal uniform crossover operator, this paper has carried out simulation experiments on SGA using the single-point crossover operator [29], two-point crossover operator [30], uniform crossover operator [31] and orthogonal uniform crossover operator. In the above four experiments, the selection operator used by the GA is a random league selection operator, the mutation operator is a uniform mutation operator, and the elite strategy is used in all of them. The population size *M* is 100, the largest number of iterations *T* is 200 times. The mutation probability $p_{m}$ is 10 %. The single-point crossover and multipoint crossover probability $p_{c}$ is 100%. The uniform crossover probability $p_{uc}$ is 70%. The content of the elite strategy of each algorithm is to replace the worst individual of the offspring with the best individual of the parent. Each algorithm is run 100 times respectively, and the average fitness of the optimal individual in each iteration of each experiment is taken to draw the simulation curve shown in Figure 3.

Figure 3 shows that the convergence rate of the uniform orthogonal crossover operator and the optimization effect are better than those of the other three kinds of common crossover operators. Compared with the other three strong crossover operators, the randomness of the uniform orthogonal crossover operator search efficiency is higher. At the same time, the orthogonal homogeneity operator can select the approximate optimal cross result of the two paired individuals through the orthogonal experiment.

Based on 100 executions of each of the above four experiments, the optimal fitness, worst fitness, average fitness, and variance of each experiment are shown in Table 3:

It can be seen from Table 3 that the optimal fitness, the worst fitness and the average fitness of the orthogonal uniform crossover operator in the results of 100 runs are better than those of the other three crossover operators, and it has the advantage of good stability.

### 4.3. Performance Simulation of a Mountain Climbing Operator Based on the Micro Genetic Algorithm

In this paper, MGA, simulated annealing algorithm (SA) [32] and Tabu search algorithm (TS) [33] are respectively applied to the standard genetic algorithm as mountain climbing operators. At the same time, the three algorithms are simulated 100 times, respectively. The three algorithms all use random league selection, uniform crossover, uniform mutation and elite strategy. The population size *M* is 100, the maximum iteration number *T* is 200, the mutation probability $p_{m}$ is 10%, and the crossover probability $p_{uc}$ is 70%. The population size of the micro genetic algorithm is 10, the maximum number of iterations is 20, the perturbation amplitude $g_{pb}$ is 10, and the mutation probability is 1%. The Tabu length of TS is 5, the maximum number of iterations is 20, and the neighborhood search method adopts uniform mutation with a mutation probability of 0.5%. SGA adopts the strategy described in the literature. The maximum number of iterations is 20, and the annealing rate is 0.01 in SA. The content of the elite strategy of each algorithm is to replace the worst individual of the offspring with the best individual of the parent.

Each algorithm is run 100 times respectively, the average fitness of the optimal individual in each iteration of each experiment is taken, and the simulation curve is drawn as shown in Figure 4.

As can be seen from Figure 4, the convergence speed and optimization effect of MGA are significantly better than for other two hill-climbing operators because MGA can search multiple directions simultaneously from the starting point, which is more efficient.

Based on executing each of the above three algorithms 100 times, the optimal fitness, worst fitness, average fitness, and variance of the optimization results of each algorithm are shown in Table 4.

In the statistical results of 100 experiments, the optimal fitness, the worst fitness and the average fitness of MGA are better than for the other two mountain climbing operators. However, for any algorithm, the performance improvement in some aspects is often offset by a performance reduction in other aspects. When MGA is applied to the standard genetic algorithm, its stability is between that of the other two algorithms. This is caused by a large number of random factors in the process of MGA.

### 4.4. HNOGA Algorithm Performance Simulation Experiment

In this paper, simulation experiments are conducted on the HNOGA algorithm, SGA, niche genetic algorithm based on scavenging mechanism (NGAC)) [34], improved ant colony optimization (IACO) [35] algorithm and greedy algorithm, and the above algorithms are run 100 times. In terms of simulation parameters, the population size *M* of the HNOGA algorithm is 100, the maximum number of iterations *T* is 200, the mutation probability $p_{m}$ is 1%, the niche neighborhood search probability $p_{m1}$ is 80%, the optimal individual neighborhood search probability $p_{m2}$ is 30%, and the number of elite individuals *q* is 30. The population size of the micro genetic algorithm is 5. The maximum number of iterations is 10. The perturbation amplitude $g_{pb}$ is 10, and the mutation probability is 1%. SGA uses random league selection, uniform crossover, uniform mutation and elite strategy. The population size *M* is 100, the maximum number of iterations *T* is 200, the mutation probability $p_{m}$ is 10%, and the crossover probability $p_{uc}$ is 70%. The content of the elite strategy is to replace the worst individual in the offspring with the best individual in the parent. NGAC uses random league selection, uniform crossover and uniform mutation and adopted the same elite strategy as the HNOGA algorithm. The population size *M* is 100, the maximum number of iterations *T* is 200, the mutation probability $p_{m1}$ is 10%, the crossover probability *q* is 70%, and the number of elite individuals is 15. In IACO, the number of ants *m* is 15. Pheromone constant *Q* is 20, pheromone factor α is 2, pheromone volatile factor ρ is 0.3, and the maximum number of iterations *T* is 200 times. The greedy algorithm is only used as the control group in this experiment, and its results are directly shown in the figure without reflecting the iterative process. Each algorithm is run 100 times, the average fitness of the optimal individual in each iteration of each experiment is taken, and the simulation curve is drawn as shown in Figure 5.

As can be seen from Figure 5, although the greedy algorithm has the advantage of a simple process, it easily falls into local optima, and the optimization effect is weaker than that of ICAO, NGAC and HNOGA. Due to the complexity of the multi-service irregular spectrum allocation problem, the optimization effect of the standard genetic algorithm is poor and the convergence speed is slow. The result of 200 iterations is only slightly better than that of the greedy algorithm. Compared with the standard genetic algorithm, the optimization effect and convergence speed of the niche genetic algorithm based on the scavenging mechanism is significantly improved due to its better population diversity. Compared with the other two genetic algorithms, the convergence speed of the HNOGA algorithm, which combines orthogonal uniform crossover, niche technology and neighborhood search of the micro genetic algorithm, is greatly improved, and the final average optimization effect has obvious advantages.

Based on executing each of the above three genetic algorithms 100 times, the optimal fitness, worst fitness, average fitness, and standard deviation of the optimization results of each algorithm are shown in Table 5.

It can be seen from Table 5 that the optimal fitness, the worst fitness and the average fitness of the HNOGA algorithm in the results of 100 runs are better than those of the other three, and it has the advantage of relatively good stability.

### 4.5. HNOGA Algorithm Priority Function Verification Simulation Experiment

In this paper, the function of the HNOGA algorithm to meet the frequency demand of high-priority devices is experimentally verified. Firstly, the verification index—the non-interference coefficient $U_{i}$—is proposed, which is defined as Equation (Equation 14): (14)Ui=∑j=1j≠iN 1−Iji·Bi+BjBi+Bj22−fic−fjcBi,fic−fjc<Bi+Bj2,i∈1,N0,fic−fjc≥Bi+Bj2,i∈1,N

As can be seen from the above equation, the non-interference coefficient $U_{i}$ represents the degree to which device *i* is not interfered with by other devices.

The HNOGA algorithm is repeated for 100 experiments, the non-interference coefficient and average value are calculated according to the above steps, and the red regular bandwidth curve in Figure 6 is drawn.

It can be seen that the degree of interference of high-priority devices is lower than that of the low-priority device, and the curve does not have any jitter, showing a strictly monotonically decreasing trend. This shows that the HNOGA algorithm meets the functional requirements of priority to meet the frequency demand of high-priority devices.

## 5. Conclusions

In the space–air–ground integrated network, the existing spectrum allocation technology cannot be applied to the multi-service scenario where the frequency devices have irregular working bandwidths and different priorities. This paper proposes a hybrid niche orthogonal genetic algorithm to solve the problem. The algorithm improves population diversity by introducing niche technology based on a clearing mechanism, proposes a new orthogonal uniform crossover operator to improve the search efficiency of the algorithm, and uses the joint optimization of the standard genetic algorithm and micro genetic algorithm to significantly improve the global searchability and local search accuracy of the algorithm. The simulation results show that the method in this paper effectively solves the problem of electromagnetic spectrum allocation of multi-service-type devices under irregular frequency demand and has the function of preferentially meeting the frequency demand of high-priority devices. In addition, compared with various existing improved genetic algorithms, the hybrid niche orthogonal genetic algorithm proposed in this paper has the performance advantages of good optimization effect, fast convergence speed and strong stability.

## Figures and Tables

**Figure 1 sensors-22-09227-f001:**
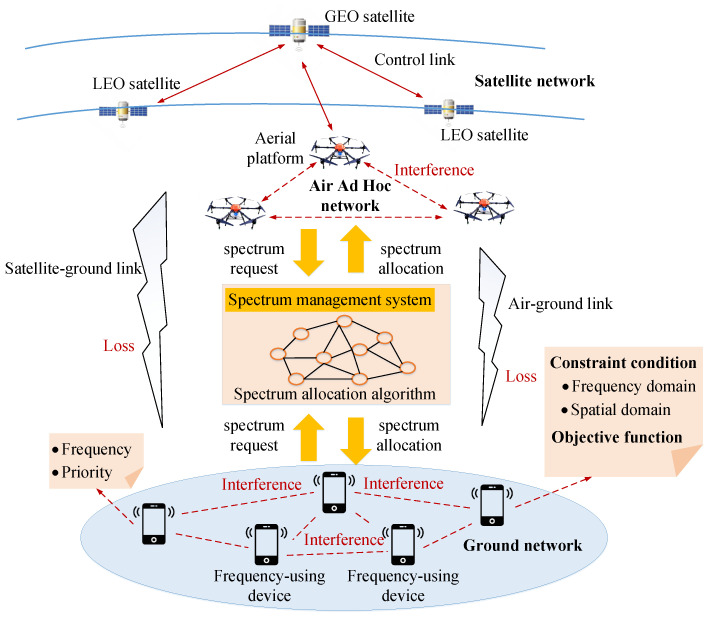
Spectrum allocation system model in space–air–ground integrated network.

**Figure 2 sensors-22-09227-f002:**
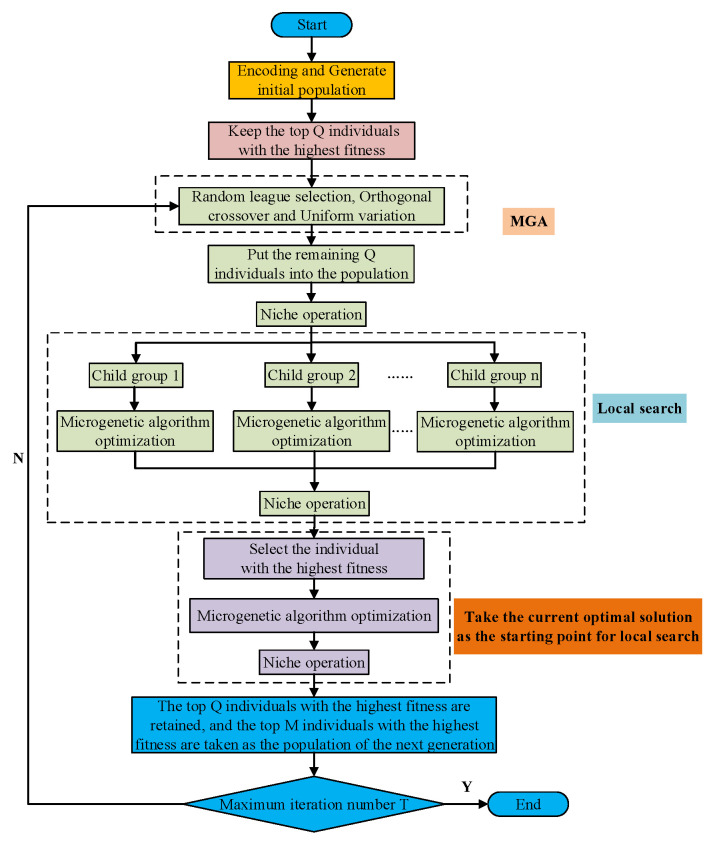
Flowchart of HNOGA algorithm.

**Figure 3 sensors-22-09227-f003:**
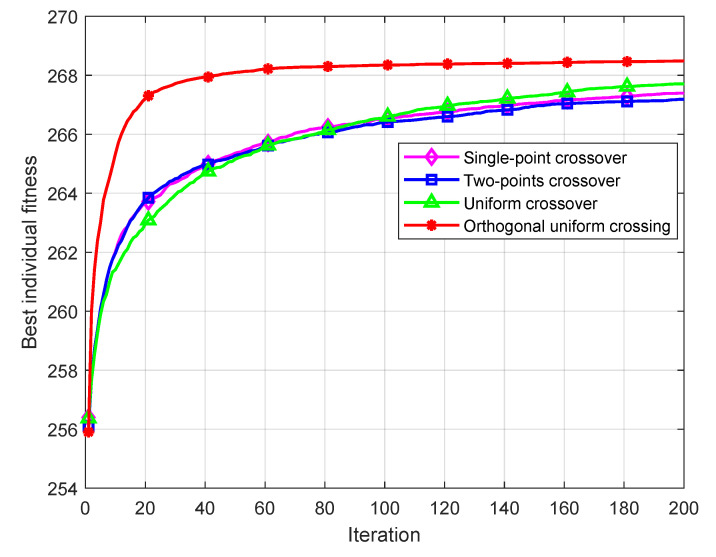
Simulation graph of crossover operator performance.

**Figure 4 sensors-22-09227-f004:**
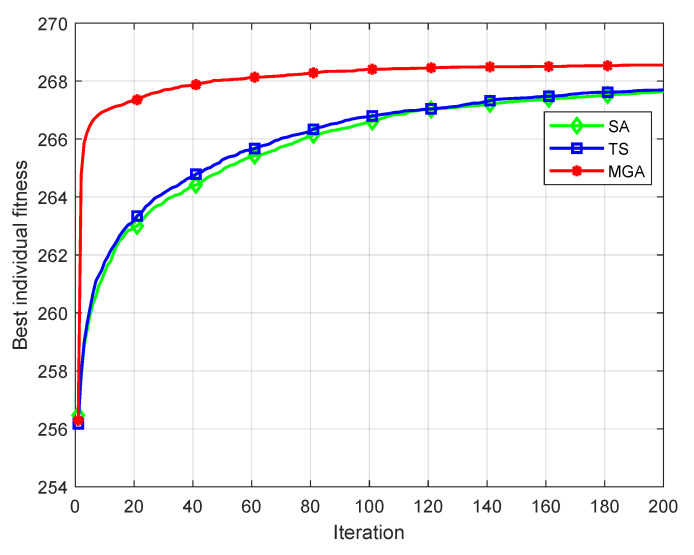
Simulation experimental curve of hill-climbing operator performance.

**Figure 5 sensors-22-09227-f005:**
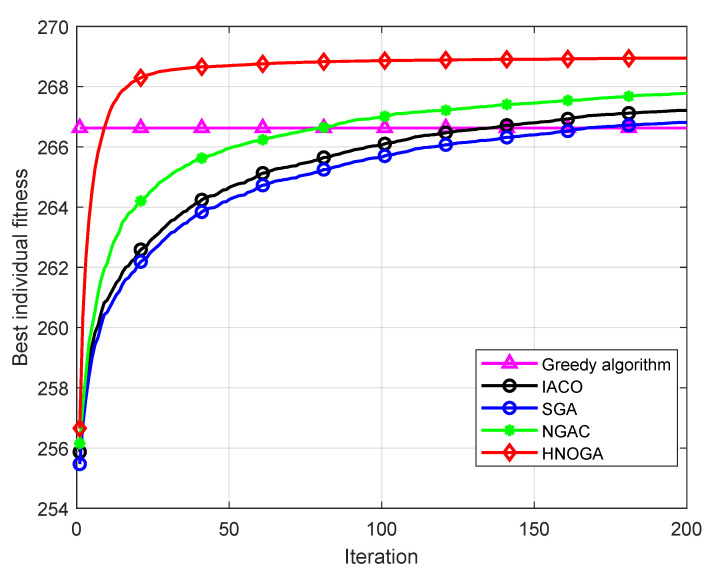
Performance simulation experimental curve of HNOGA algorithm.

**Figure 6 sensors-22-09227-f006:**
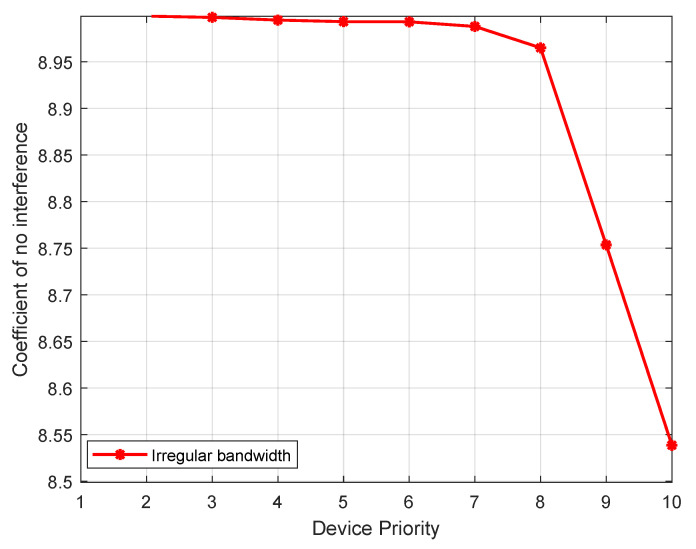
Experimental curve of priority function verification.

**Table 1 sensors-22-09227-t001:** Device parameters.

Priority	Business Type	Receiving Sensitivity	Geographical Position	Inherent Frequency	Working Frequency	Working Bandwidth	Transmission Power
$p_{k}$	$T_{yk}$	$S_{k}$	$\left(x_{k},y_{k},z_{k}\right)$	$\left(f_{kgs},f_{kge}\right)$	$\left(f_{ks},f_{kg}\right)$	$B_{k}$	$P_{k}$

**Table 2 sensors-22-09227-t002:** Device parameters.

Device ID	Priority	Location (Height, Latitude, Longitude)	Sensitivity (dBm)	Pt (dBm)	Inherent Frequency (MHz)	Bandwidth (MHz)
1	2	$\left(30,35,4\right)$	−93	14	$\left(0,220\right)$	20
2	0	$\left(31,35,4\right)$	−86	15	$\left(0,220\right)$	35
3	4	$\left(32,35,4\right)$	−89	16	$\left(0,220\right)$	22
4	3	$\left(33,35,4\right)$	−90	14	$\left(0,220\right)$	16
5	1	$\left(34,35,4\right)$	−88	15	$\left(0,220\right)$	21
6	2	$\left(35,35,4\right)$	−87	12	$\left(0,220\right)$	18
7	0	$\left(36,35,4\right)$	−95	13	$\left(0,220\right)$	14
8	4	$\left(37,35,4\right)$	−94	12	$\left(0,220\right)$	23
9	3	$\left(38,35,4\right)$	−90	19	$\left(0,220\right)$	12
10	1	$\left(39,35,4\right)$	−88	17	$\left(0,220\right)$	20

**Table 3 sensors-22-09227-t003:** Experimental results of crossover operator performance verification.

	Optimal Fitness	Worst Possible Fitness	Average of Fitness	Variance
Single-point crossover	268.865	265.71	267.398	0.382667
Two-point crossover	268.533	265.165	267.189	0.489417
Uniform cross	268.799	265.904	267.711	0.3326
Orthogonal uniform crossing	269.244	266.742	268.489	0.204459

**Table 4 sensors-22-09227-t004:** Experimental results of crossover operator performance verification.

	Optimal Fitness	Worst Possible Fitness	Average of Fitness	Variance
SA	269.169	265.953	267.635	0.419217
TS	268.991	266.11	267.696	0.354487
MGA	269.337	267.321	268.555	0.150796

**Table 5 sensors-22-09227-t005:** Experimental results of crossover operator performance verification.

	Optimal Fitness	Worst Possible Fitness	Average of Fitness	Variance
SGA	268.799	265.904	267.711	0.3326
IACO	268.698	266.015	267.743	0.313765
NGAC	268.622	266.127	267.786	0.275283
HNOGA	269.422	268.201	268.946	0.0733335

## Data Availability

Not applicable.

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
