# Peer review of "Electromagnetic Spectrum Allocation Method for Multi-Service Irregular Frequency-Using Devices in the Space–Air–Ground Integrated Network"

_sensors, 2022, doi:10.3390/s22239227_

Round 1

Reviewer 1 Report

The paper needs some major improvements before publication:

·      The manuscript needs to be carefully checked because there are many grammatical errors and typos.

·      There are quantities which are not properly defined such as working frequency in Table 1 and equation 7.

·      There must be a reference cited for equation 5.

·      The details of the algorithm are not very clear. Please add more details about the proposed algorithm such as illustration of coding and decoding structures.

·      There is a classification algorithm called RUS-boosted trees that was developed for an imbalanced data set. Why this algorithm is not discussed in the paper?

·      It is better to add a section discussed the complexity of the proposed algorithm compared with the existing algorithm.

Reviewer 3 Report

This paper proposes a spectrum allocation method based on hybrid niche orthogonal genetic algorithm for multi service irregular frequency equipment is proposed, which well solves the spectrum allocation problem of multi service equipment in the space-air-ground integrated network under different frequency-using requirements. The presentation of this paper is well, I have some specific remarks/questions on this work. There are few points that can be addressed to improve the manuscript.

1.    In Section 2.3, in the process of building the objective function, whether it is necessary to consider the conflict of each device in the time domain;

2.    In subsection 3.2.1, the rules of HNOGA algorithm proposed in this paper for real number coding to map chromosome arrays need to be further elaborated;

3.    In subsection 3.2.3, whether the operation steps of the selection operator should be given;

4.    I have noticed that a uniform crossover operator based on orthogonal experiment is proposed in this paper. Please explain the rationality of adopting orthogonal experiment;

5.    I have noticed that the HNOGA algorithm proposed in this paper uses the niche technology based on the Cleaning Procedure. Please elaborate on how the niche technology can avoid the algorithm from falling into the local optimal solution.

Round 2

Reviewer 2 Report

The authors have addressed most of my concerns, however, it is suggested to introduce the following recent works in space-air-ground integrated network field [R1]-[R2] to highlight the state-of-the-art of this paper.

[R1] “Joint beamforming and power allocation for satellite-terrestrial integrated networks with non-orthogonal multiple access,” IEEE Journal of Selected Topics in Signal Processing, vol. 13, no. 3, pp. 657-670, June 2019.

[R2] “Supporting IoT with rate-splitting multiple access in satellite and aerial-integrated networks,” IEEE Internet of Things Journal, vol. 8, no. 14, pp. 11123-11134, Jul. 2021.

Reviewer 3 Report

This version  can be accepted. No further comments.
